# An Evolutionary Model of Progression to AIDS

**DOI:** 10.3390/microorganisms8111714

**Published:** 2020-10-31

**Authors:** Igor M. Rouzine

**Affiliations:** Laboratory of Computational and Quantitative Biology, 7238 CNRS-UPMC, Institut Biologie Paris-Seine, Sorbonne Université, Campus Pierre et Marie Curie, 75005 Paris, France; igor.rouzine@sorbonne-universite.fr

**Keywords:** mathematical model, virus adaptation, host-virus interaction, progression to AIDS, homeostasis

## Abstract

The time to the onset of AIDS symptoms in an HIV infected individual is known to correlate inversely with viremia and the level of immune activation. The correlation exists against the background of strong individual fluctuations demonstrating the existence of hidden variables depending on patient and virus parameters. At the moment, prognosis of the time to AIDS based on patient parameters is not possible. In addition, it is of paramount importance to understand the reason of progression to AIDS in untreated patients to be able to learn to control it by means other than anti-retroviral therapy. Here we develop a mechanistic mathematical model to predict the speed of progression to AIDS in individual untreated patients and patients treated with suboptimal therapy, based on a single-time measurement of several virological and immunological parameters. We show that the gradual increase in virus fitness during a chronic infection causes slow gradual depletion of CD4 T cells. Using the existing evolution models of HIV, we obtain general expressions predicting the time to the onset of AIDS symptoms in terms of the patient parameters, for low-viremia and high-viremia patients separately. We show that the evolution model of AIDS fits the existing data on virus-time correlations better than the alternative model of the deregulation of homeostatic response.

## 1. Introduction

Almost all (99.5%) people infected with HIV develop the acquired immunodeficiency syndrome [1,2,3]. The virus is usually isolated from men who have unprotected sex with men, sex workers, and intravenous drug users [4]. The disease manifests itself in a broad range of opportunistic infections and cancers in previously healthy individuals including pneumonia, Kaposi’s sarcoma, Mycobacterium tuberculosis and many other infections [5,6,7,8]. AIDS patients display an impaired immune response, which includes low lymphocyte proliferative responses ex vivo and depletion of T-helper cells [5,6,7]. HIV selectively infects CD4+ T cells and macrophages and depletes CD4+ T cells, which in fact is coincident with the onset of AIDS symptoms [9,10]. Suppression of HIV replication with antiretroviral therapy (ART) rapidly increases peripheral blood CD4+ T-cell counts and restores immune function [11,12]. These results show that the immunodeficiency in untreated individuals is caused by the decrease in the CD4+ T cell count and the loss of CD4+ T-cell function, which impairs T cell immunity to pathogens and causes the onset of symptoms within 2–20 years. The effect is reproduced experimentally in nonhuman primates where infection with chimeric simian/human immunodeficiency viruses leads to rapid CD4 T cell depletion, AIDS, and death within 1–2 years [13,14]. 

Progression to AIDS does not occur due to a slow consumption of pre-existing cells by the virus. HIV infection represents a steady state with a rapid turnover of infected cells (1 day) where the death of infected cells in about 1 day is compensated by new rounds of infection [11,12]. Cells targeted by HIV memory CD4+ CCR5+ T cells are also continuously replenished, but with a longer average lifespan of several weeks [15,16,17]. The steady state is almost exact, but not quite, with a slow decrease in CD4 T cell count in time. Before this quasi-steady state sets up, target cells are deeply depleted during the virus peak of the acute phase of infection [18,19,20,21]. The time to AIDS inversely correlates with both the virus load [22,23] and the level of immune activation across HIV-infected subjects [23,24,25,26].

The existence of an exact steady state can be explained using simple mathematical models of hunter-prey type [27,28]. However, the reasons for the slow change in that state, eventually causing AIDS on time scales much longer than the lifespans of both target and infected cells, remain unclear and must be additional to the factors responsible for an exact steady state. The existence of non-pathogenic simian immunodeficiency virus strains, which do not cause gradual depletion of CD4 T cells and progression to AIDS in their natural hosts, confirms this view [29,30,31,32,33]. A natural assumption is that these hosts and viruses are mutually adapted, while pathogenic infections occur after cross-species transfer and are not adapted yet. 

We will exploit the last idea in the present manuscript, by presenting a model of CD4 T cell depletion driven by virus evolution adapting to a new host. The rapid evolution and high diversity of HIV in an infected host is well-documented. The speed of virus adaptation to a host is predicted by a family of mathematical models, including the cases of asexual evolution at low MOI [34,35,36,37,38,39,40], evolution with recombination at high MOI [41,42,43,44,45,46], evolution forced by the immune response [41,47,48,49], and the effect of epistasis [50,51,52,53,54,55]. We will combine the results for the adaptation speed for several simple models of evolution with a simple immunological model to predict the speed of CD4 T cell depletion.

As we mentioned, the time of the onset of AIDS symptoms, due to gradual CD4 T cell depletion, varies broadly among untreated individuals. The progression time is shown to correlate negatively with the virus load and the degree of immune activation [22,23,24,25,26]. However, the correlation exists against the background of strong individual fluctuations, demonstrating the existence of hidden parameters depending on a patient and a virus variant. At the moment, making an accurate prognosis for individuals is impossible, because the cause of the slow transition to AIDS is unknown. In the present work, we will show how to combine the existing models of virus evolution and immune response with data from patients to identify these hidden parameters and make prognosis for individual patients.

We propose a working model postulating that the slide towards AIDS is caused by virus evolution. An alternative hypothesis of progression to AIDS, consistent with a body of data but not implemented as a closed mathematical model, postulates that the deregulation of a host homeostatic mechanism, and the disruption of immune tissue microenvironment by the virus are responsible for the depletion of CD4 T cells (see [56] for a review). In the present work, we will contrast predictions of the two alternative models (Figure 1) and compare them with correlation data. Thus, the present work offers the first method of the accurate prognosis of progression to AIDS based on the combination of evolutionary and immunological models and patient data.

## 2. Materials and Methods 

Before introducing factors responsible for CD4 T cell depletion, we introduce the two simplest models of virus dynamics predicting an exact balance between the death and replenishment of viremia and target cells. The first model does not include the immune response, and the second includes it. Then, we will take into account the genetic evolution of the virus. We will consider alternative models of progression to AIDS based on homeostasis deregulation in the Discussion.

### 2.1. Model of Virus Dynamics in the Absence of the Immune Response 

The first model describes virus dynamics in the absence of immune response and is appropriate for very small children (1 year and younger). The standard model includes uninfected target cells and infected cells and describes their dynamics, as follows: (1)$$\frac{dT}{dt}=\lambda -\;\left(d_{T}+kI\right)\;T$$
(2)$$\frac{dI}{dt}=kIT-d_{I}I$$
where T  denotes the number of uninfected CD4 CCR5 T cells, I is the number of the infected cells, λ is the linear rate of homeostatic replenishment of effector CD4 T cells from precursors cells assumed to be constant. Next, $d_{T}$ is the natural death rate of target cells, $d_{I}$ is the death-rate of infected cells, and k is the virus infectivity parameter. Please note that we do not include the virion number explicitly, because virions are very short lived in vivo, about 6 h, and their number is proportional, at any time, to the number of infected cells. 

### 2.2. Model of Virus Dynamics with the Cytotoxic Immune Response 

The simplest model including the cytotoxic immune response is a simplified version of the model used in [57], as follows:(3)$$\frac{dT}{dt}=\lambda -\;\left(d_{T}+kI\right)\;T$$
(4)$$\frac{dI}{dt}=kIT-d_{I}I\left(1+\frac{E}{E_{0}}\right)$$
(5)$$\frac{dE}{dt}=aE\frac{I}{I+I_{0}}-d_{E}E$$

Here, Equation (3) is the same as Equation (1), while in Equation (4), as compared to Equation (2), the death-rate of infected cells is modified to include the killing by the immune effector cells whose number is denoted by E, and $E_{0}$ is the number of immune cells that increases the death rate by the factor of two. Equation (5) describes the dynamic of the effector cells E, where a is the maximum proliferation rate of effector CD4 T cells (lumped here together with their immature precursors), $I_{0}$ is the characteristic infected cell number at which half-maximum immune cell expansion is attained, and $d_{E}$ is the natural death rate of effector cells. Note that the model does not include helper-dependent CTL immune response [58,59] important in lowest-viremia patients and virus controllers. In the present work, we will focus on high-viremia to moderately-low-viremia patients and neglect the helper-cell response. 

### 2.3. The Rate of Adaptation to a Host 

The virus adaptation rate, which is the rate at which Maltusian fitness changes per unit time, $V=d\log\;R_{0}/dt,$ where $R_{0}$ is the basic reproduction number, is evaluated based on a model of evolution. In the past, the value of V  has been expressed, in the general form, in terms of the evolutionary model parameters for various models [34,35,39,42,44,60,61,62]. In the basic scenario when fitness benefit of a mutation is fixed, and in the absence of recombination [34], the adaptation rate is given by
(6)$$V=2s^{2}\frac{\log\left(I_{ss}\sqrt{sU_{b}}\right)}{\log^{2}\frac{s\;\log\;I_{ss}}{U_{b}}}$$
where $U_{b}$ is the rate of beneficial mutations per genome per replication cycle, s is the fitness gain (increase in log $R_{0}$) per mutation, which is assumed to be the same for all loci (SNP), and $I_{ss}$ is the steady state value of the number of infected cells obtained from the condition that the right-hand side of each of Equations (1) and (2) or Equations (3)–(5) is equal to zero. To be valid, the arguments of both logarithms in Equation (6) must be much larger than the unit, which implies that $I_{ss}$ cannot be too small. 

In the realistic case of a distributed selection coefficient with a distribution density—for example, $\rho \left(s\right)=\left(1/s_{av}\right)\;\exp\left(-s/s_{av}\right)$, often seen in experiment [63,64,65,66,67]—we have a more complex formula for the adaptation rate, V. However, at sufficiently large values $I_{ss}U_{b},$ it can be mapped to the constant-*s* case in Equation (6), using the effective parameters [39]
$$s_{\mathrm{eff}}=s^{*},\;U_{\mathrm{eff}}\;\sim \;U_{b}\sqrt{2\pi V}\;\rho \left(s^{*}\right)$$
where V is found by substituting the second equation into Equation (6) and solving it by iterations, which converge after one or two iterations. The specific expression for the effective selection coefficient, $s^{*},$ derived analytically in [39] in terms of the model parameters, is not essential for our aim. We can simply use the model with a constant selection coefficient, $s_{\mathrm{eff}},\;$ and a beneficial mutation rate, $U_{b}$, replaced with $U_{\mathrm{eff}}$ in Equation (6). The value of $s_{\mathrm{eff}}$ is estimated directly from patient sequence data, as explained below.

Evolution is asexual when the virus load is modest. In patients with a high viremia $>10^{5}$ RNA copy/mL, we need to take into account recombination. The effective value of outcrossing rate, r, depends on MOI and is estimated directly from patient sequence data, as explained below. In an average patient with viremia $3\times 10^{5}$, recombination accelerates the adaptation speed, V, by the factor of 2 [60]. In this case, the expression for the adaptation rate, V, in the presence of recombination and mutation has a form [46]
$$V=\{\begin{array}{c} 2r^{2}\frac{\log\;(I_{ss}U_{b})}{\log^{2}\left(r/s\right)}\;1\ll {\left(r/s\right)}^{2}\ll \frac{I_{ss}U_{b}}{\log\left(I_{ss}U_{b}\right)}\\ I_{ss}U_{b}s^{2}\left(1-\frac{4I_{ss}U_{b}s^{2}}{r^{2}}+\ldots \right)\;\;{\left(r/s\right)}^{2}\gg \frac{I_{ss}U_{b}}{\log\left(I_{ss}U_{b}\right)} \end{array}$$
where strong inequality $I_{ss}U_{b}\gg 1\;$ is assumed (and is valid for all but extremely-low viremia patients). The first line describes the effect of recombination helping new beneficial alleles to be established in a population. The second line corresponds to the high MOI and hence high recombination rates, r, such that linkage effects are mosty compensated by recombination, and the adaptation at different loci is independent. As we mentioned, with the decrease in viremia below $10^{5}\;$ RNA copy/mL, the adaptation rate crosses over to a much smaller, asexual result given by Equation (6).

### 2.4. Experimental Samples 

To use this method, patients need to be classified according to their therapy status, the progressor or controller status, the age, the biological sex, and the infecting HIV subtype. The results should be used for each cohort separately. A one-time sample of 50 mL blood, 5 mL per parameter, would be sufficient to detect all parameters, the most critical being the virus load, selection coefficient, s,  and recombination rate, r. The representative time point of AIDS is defined per CDC as the time when CD4 count falls below 200 CD4 cell/mm^3^, at which point most patients develop AIDS symptoms. This prediction can be easily updated for any other chosen threshold, depending on the type of symptoms—for example, 350 or 50 CD4 cell/mm^3^. 

The standard error of the prediction of $t_{\mathrm{AIDS}}$ is determined from the experimental error of each measured input parameter, denoted by X, and the partial derivative in that parameter, $\partial t_{\mathrm{AIDS}}/\partial X$, calculated from Equation (6). The most critical and sensitive parameters are the number of the infected cells, the selection coefficient, and the recombination rate, which are estimated from RNA virus sequences isolated from a patient. We will discuss parameters in the Results section.

## 3. Results

Below, we connect together the models of immunology and evolution to predict the time of progression to AIDS from individual patient’s data.

### 3.1. Exact Steady State in the Absence of the Immune Response 

From Equations (1) and (2), by putting the time derivatives to zero, we obtain an exact steady state with the levels for uninfected CD4 T cells, $T_{ss}$, and infected CD4 T cells, $I_{ss}$
(7)$$T_{ss}=\frac{T_{0}}{R_{0}}$$
(8)$$I_{ss}=\frac{d_{T}}{k}\left(R_{0}-1\right)$$

Here, $T_{0}$ denotes the normal CD4 cell number in uninfected patients (at I = 0), and $R_{0}$ is the basic reproduction number
(9)$$T_{0}\equiv \frac{\lambda }{d_{T}},\;R_{0}\equiv \frac{k\lambda }{d_{I}d_{T}}$$Note that, for an infection to occur, $R_{0}$ has to be larger than 1. 

### 3.2. Exact Steady State in the Presence of CD8 T Cell Immune Response 

By setting the right-hand side of Equations (3)–(5) to zero, we obtain the steady state numbers for $T_{ss}$, $I_{ss}$, and CD8 T cell effector cells, $E_{ss}$
(10)$$\begin{array}{c} T_{ss}=\frac{T_{0}}{1+\frac{R_{0}}{R^{*}}},\;R^{*}\;\equiv \frac{T_{0}d_{T}}{I_{ss}d_{I}}\;\gg \;1\;\\ \;I_{ss}=\frac{I_{0}d_{E}}{a-d_{E}}\\ E_{ss}=E_{0}\left[\frac{R_{0}}{1+\frac{kd_{E}I_{0}}{d_{T}\left(a-d_{E}\right)}}-1\right] \end{array}$$

Here, the CD4 count in uninfected patients, $T_{0}$, and the basic reproduction number, $\;R_{0}$, are given by Equation (9). Here we neglect the existence of latent cells and do not consider the early mucosal phase of infection, only the systemic phase of infection [57]. Note that the model does not include the helper-dependent CTL immune response important in lowest-viremia patients [58,59]. In the present work, we will focus on high-viremia and medium-viremia patients and neglect the helper-cell response.

### 3.3. Slow Progression to AIDS

According to our main hypothesis (alternative hypothesis are discussed below in the Discussion), the steady state value of CD4 T cells, $T_{ss}$ (Equation (7) or Equation (10)), is modulated by the basic reproduction number $R_{0}$, which increases slowly in time, as the virus slowly evolves, genetically adapting to a patient. The virological reason for the increase in $R_{0}$ is the increase in the infectivity per infected cell, k, which includes both the average number of virions produced by an infected cell and infectivity per virion. Note that, if parameter $R_{0}$ changes very slowly in time, the system is in a quasi steady state at each moment of time, with a slowly changing count of CD4 T cells. This explains right away why the number of CD4 T cells, $T_{ss},$ slowly decreases over the course of HIV infection, leading to AIDS. 

Early AIDS symptoms start below $T_{ss}\left(t_{\mathrm{AIDS}}\right)$
= 200 cell/mkl blood (some sources give 350), whereas the average initial CD4 T cell level in a chronic HIV infection following the acute phase of infection is $T_{ss}\left(0\right)$
= 500 cell/mkl blood. Our models explain this decline of $T_{ss}\left(t\right)$ through Equations (7) and (10) and the fact that $R_{0}\left(t\right)$ increases in time. The time to the onset of symptoms can be calculated, as follows
(11)$$t_{\mathrm{AIDS}}=\frac{\log\;R_{0}\left(t_{\mathrm{AIDS}}\right)-\log\;R_{0}\left(0\right)}{\frac{d\;\log\;R_{0}}{dt}}\;$$
which assumes that $V=\frac{d\;\log\;R_{0}}{dt}\;$ is constant in time. Using Equation (7) for $T_{ss}$ for the model without the immune response, we can calculate the time to progression to AIDS as
(12)$$t_{\mathrm{AIDS}}=\psi /V,\;\psi \equiv \log\frac{T_{ss}\left(0\right)}{T_{ss}\left(t_{\mathrm{AIDS}}\right)}\;$$
where $V\equiv \;d\;\log R_{0}/dt\;$ is the virus fitness increase rate that has to be found from a model of virus evolution described in the Methods (Section 2.3). Note that ψ  is close to 1 on average but may vary between patients. 

In the presence of the immune response, from Equation (10) for $T_{ss}$ and Equation (11), we get the new estimate for the time of progression to AIDS
(13)$$t_{\mathrm{AIDS}}=\frac{1}{V}\;\left\{\psi +\log\left[1+\frac{R^{*}}{R_{0}\left(0\right)}\left(1-e^{-\psi }\right)\right]\right\}\;$$
which is several-fold longer than in the absence of the immune response, Equation (12), and ψ≈1. This is consistent with the observation that the average time to AIDS in untreated adults, 10 years, is much longer than in infants, about 1–2 years. In Equations (12) and (13), the virus adaptation rate, V, has to be evaluated based on a model of evolution [see *Methods*, Equation (6) and below]. 

### 3.4. Explaining the Observed Negative Correlation between the Time to AIDS and Virus Load and Immune Activation 

The time to AIDS predicted by Equation (13) can fit the negative correlation between the time to AIDS and the virus load using two composite fitting parameters (Figure 2, red lines; see also Section 4 below). Indeed, the virus load, v, is known to be proportional to the total number of infected cells, $I_{ss},$ so that an average untreated patient with $v=3\times 10^{4}\;$ RNA/copy ml has $\sim 3\times 10^{8}\;$ infected cells, so we have unit conversion between the number of cells and viremia, $I_{ss}=v\theta ,\;$ where $\;\theta \;\approx 10^{4}\;\mathrm{cell}\cdot \mathrm{ml}/\mathrm{RNA}\;\mathrm{copy}$.

Our prediction for the time to AIDS as a function of virus load in adults, Equation (13), can be presented in the form with three composite parameters, as follows:(14)$$t_{\mathrm{AIDS}}=\frac{A}{\log\left(Bv\right)}\left[1+\log\left(1+\frac{C}{v}\right)\right]\;$$
which is obtained by expressing V and $R^{*}$ in terms of $I_{ss}\;$ from Equations (6) and (10), respectively, and substituting them into Equation (13). Here
$$A=\frac{1}{s^{2}}\log^{2}\frac{s\;}{U_{b}}\;B=\theta v\sqrt{sU_{b}}$$
are composite fitting parameters adjusted to fit data, and
$$C=\frac{T_{0}d_{T}}{\theta d_{I}R_{0}\left(0\right)}\left(1-e^{-1}\right)$$
is fixed at any large value, $C>10^{6}\;\mathrm{RNA}\;\mathrm{copy}/\mathrm{mL}.$ We fit this expression to data in Figure 2 using the least-squares method implemented in MATLAB^TM^ in the program fminsearch.m. Manual fitting is used for the starting point. Please note that we do not use experimental estimates of parameters from Table 1, which is designed solely for patient prognosis. Fitting is done in terms of the two composite parameters, A, B. We have checked that the variation in the only fixed parameter, C, has a negligible effect on the fitting accuracy.

The evolution model also explains, qualitatively, the positive correlation of the speed of the AIDS progression with the level of the immune activation [23,24,25,26]. Indeed, adaptation of HIV is driven by the early immune escape in epitopes and subsequent epistatic compensatory mutations [41]. The average selection coefficient s and the number of highly diverse sites correlate with the intensity and the breadth of the T cell immune response, of which the cytokine level is a marker. Because, at low virus loads (asexual regime) and very high virus loads, the maximum adaptation rate is proportional to the square of selection coefficient, Equation (6), this fact explains the correlations.

### 3.5. Parameter Sensitivity for AIDS Prognosis in Individual Patients 

This method can be useful for patient prognosis if we know the input parameters in Equation (13). They can obtained, in principle, from the virological data from individual patients, which can be measured or inferred, as follows:
Coefficient ψ  is given by Equation (12), where $T_{ss}\left(0\right)$ is the current CD4 cell count in the patient that is usually known.Total number of infected cells, $I_{ss}$, can be estimated from viremia in RNA copy/mL blood measured by qPCR using a standard unit converter [42]. As a rule of thumb, 10,000 productively infected cells in the body roughly correspond to 1 RNA copy/mL [68].Mutation rate, $U_{b}=\mu L$, where the average mutation rate per site, $\mu =3\times 10^{-5},\;$is known [69], and L  is the number of highly variable sites (SNP) in a virus strain in a patient, which can be measured directly.Effective selection coefficient, s, and outcrossing rate, r. Two methods of estimating them from patient sequences were developed, and we refer the reader to this work [45,60]. The first, more accurate method [60] is based on Monte-Carlo simulation and comparison of two measures of linkage disequilibrium calculated for simulated sequences with those extracted from real virus sequences at one time. The second, simpler method [45] uses two additional approximations and two time points. For an average patient, they produce similar estimates, and we recommend using both, to be on the safe side. These are not the only existing methods of estimating s, but they must suffice.Input parameters $T_{0},\;\;d_{T},\;$ and $\;d_{I}$ vary little between patients, and the respective values are in the argument of a log function. Hence, the inter-patients variability does not affect results and these values can be replaced with their well-known averages over patients. In principle, for better accuracy, they can also be measured directly for each individual [59].Basic reproduction ratio, $R_{0}\left(0\right),\;$ has to be measured in the primary infection and is typically unknown for a chronic patient. Its value enters the argument of a large logarithm in Equation (13), which means that it can be approximated by its average over patients, $\langle R_{0}(0)\rangle =\;$8 [70]. 


The relative sensitivity of $t_{\mathrm{AIDS}}$ to patient parameters can be conveniently measured from the relative experimental errors by calculating double log derivatives of Equation (13), as follows:$$\frac{\partial \;\log\;t_{\mathrm{AIDS}}}{\partial \;\log\;s}=-2\;\frac{\partial \;\log\;t_{\mathrm{AIDS}}}{\partial \;\log\;\log\;I_{ss}}=-1\;\frac{\partial \;\log\;t_{\mathrm{AIDS}}}{\partial \;\log\;R_{0}}=-\frac{\partial \;\log\;t_{\mathrm{AIDS}}}{\partial \;\log\;R^{*}}=-\frac{D}{\left(1+D\right)\left(\psi +\log\left(1+D\right)\right)}\;D\equiv \frac{R^{*}}{R_{0}}\left(1-e^{-\psi }\right)$$

Suppose we have the 95% confidence interval for $\log\;X_{i}$ from an experiment, denoted by $\varepsilon_{i},$ where $X_{i}$ is any of the above four parameters: $s,\;\log\;I_{ss},\;R_{0},\;$ and $\;R^{*}.$ Then, we can estimate the 95% confidence interval for log $t_{\mathrm{AIDS}},$ denoted by ε, as
$$\varepsilon =\sqrt{\overset{4}{\underset{i=1}{\sum }}{\left[\frac{\partial \;\log\;t_{\mathrm{AIDS}}}{\partial \;\log\;X_{i}}\right]}^{2}\varepsilon_{i}{}^{2}}$$

Here the experimental error of $\log\;R^{*}\;=\log\;\frac{T_{0}d_{T}}{I_{ss}d_{I}}$ is expressed in terms of the experimental errors of the measured values of logarithms of $T_{0},\;d_{T},\;I_{ss},\;d_{I}$ using the same formula with the double log derivative in the brackets equal to ±1.

## 4. Discussion

We have derived general expressions predicting the time to AIDS from the rate of virus adaptation measured genetically and causing depletion of target cells. Input parameters can be estimated from single-time or two-time blood sample, as we have described. Now we need to consider a very popular alternative model, in which progression to AIDS is determined by deregulation of homeostasis [56], and compare its predictions with patient data to find out which model works better. Enhanced activation of T cells has been implicated in a faster disease progression [24,25,26], but this effect does not lead to a predictive model of AIDS progression [71]. The diversity threshold model has the same problem and it has received serious critique [71]. A plausible alternative explanation of progression to AIDS, based on data, is the deregulation of homeostasis [56], which we test now.

Because we now focus on homeostatic replenishment, we need to make this part of the model more accurate. Above, we assumed a fixed linear replenishment rate, λ. In fact, homeostatic mechanism shuts down replenishment when T increases to a threshold
(15)$$\lambda \left(T\right)=\lambda_{0}\left(1-\frac{T}{T_{0}}\right)$$
where $T_{0}$ and $\lambda_{0}\;\mathrm{are}$ now two independent constant parameters. Suppose the maximal number of divisions is finite due, for example, to the telomere length restriction—i.e., a possible number of new precursor cells has a finite limit, Λ, as given by
$$\lambda \left(T_{ss}\right)t_{\mathrm{AIDS}}=\Lambda$$

Substituting Equation (15) into Equation (3) at steady state, from the above equation, we get
(16)$$t_{\mathrm{AIDS}}=\frac{\Lambda }{\lambda_{0}}\frac{(\lambda_{0}/T_{0})+kI_{ss}}{kI_{ss}}\equiv a\frac{b+I_{ss}}{I_{ss}}\;$$
where we assumed $kI_{ss}\gg d_{T},\;$ or $R_{0}\gg R^{*},\;$ which is experimentally correct for an average untreated infected person, where the ratio is the turnover time ratio of T cells in infected and uninfected patients, ~3−6 [15,16,17]. Here, a, b  are new composite parameters, which do not depend on the virus load, $I_{ss}.$

Fitting the last expression for $t_{\mathrm{AIDS}}\;$ to data on the negative correlation between virus load and the time to AIDS by using two fitting parameters, a and b, to patient data in [22] predicts a curve, comprising a steep decline crossing over to a plateau (Figure 2, cyan curve). Such a curvature is not seen in the data, which form a straight stripe in the double log scale. 

Now, let us change the version of this model and assume that, instead of limited division, the progress to AIDS is caused by the decay of homeostatic environment due to the accumulation of virus products [56]. This process can be included into our basic model by postulating that the virus gradually erodes the maximal homeostatic source intensity, $\lambda_{0}$, which thus slowly decays in time. Formally, we have
(17)$$\frac{d\lambda_{0}}{dt}=-\alpha I_{ss}\lambda_{0}\left(t\right)$$

After solving this simple equation, we obtain an exponential decay with the rate depending on the virus load
$$\lambda_{0}\left(t\right)=\;\lambda_{0}\left(0\right)e^{-\alpha It}$$

Using the condition determining the onset of AIDS
$$T_{ss}\left(t_{\mathrm{AIDS}}\right)=T_{\mathrm{AIDS}}$$
and inserting $\lambda_{0}\left(t\right)\;$ instead of  λ  into Equations (9) and (10), we get
(18)$$t_{AIDS}=\frac{1}{\alpha I_{ss}}\log\left(\frac{\lambda_{0\;}\left(0\right)}{kI_{ss}T_{\mathrm{AIDS}}}\right)\equiv \frac{1}{\alpha I_{ss}}\log\frac{\beta }{I_{ss}}$$
where we assumed $kI_{ss}\gg d_{T}$ again, and β is a composite parameter which does not depend on $I_{ss}$. We fit this expression to the same data using, again, two fitting parameters, α  and  β. We observe that the best-fit curve for $t_{\mathrm{AIDS}}\;$ decreases far too rapidly with $I_{ss}\;$ compared to the actual correlation (Figure 2, blue line). This model can be excluded decisively.

In contrast, the prediction of our virus evolution model, Equation (6), also using two composite fitting parameters, fits the observed slow correlation in a broad range of values rather easily (red lines, Figure 2). The reason for the good fit is the logarithmic dependence of the evolution rate on the population size of infected cells. This is a universal property of various models with clonal interference and multiple evolving sites cited in the Methods and has no relation to the particular choice of the model (dominant factors) of the immune response.

Of course, this is not the end of comparison of our model with alternative models, but only the beginning. For example, these models can be compared further using the longitudinal samples of CD4 T cell counts obtained from patients. For the virus adaptation model, we can predict the CD4 count decline slope, $dT_{ss}/dt,\;$ from Equation (10) and Equation (6) and compare it with the observed decline. Note that the limited-division model above predicts an abrupt drop in CD4 T cell count before the onset of AIDS, instead of the gradual decline usually observed in patients. At the present moment, we can conclude that various versions of the evolution model work better than the versions of the homeostatic model. Its better performance is due to a very slow, logarithmic dependence of the speed of virus evolution on the virus load. The slow dependence of evolution rate on population size is a phenomenon universal for asexual (or weakly sexual) populations (Section 2.3). In a homeostasis-based model of progression, such a slow dependence is much harder to explain. 

We have considered two simplest models of the immune response. The models do not include the helper-dependent CD8 T cell immune response [58,59] important in patients with very low viremia. In the present work, we have focused on moderate to high viremia patients and hence neglected the helper-cell response. The models of the immune response may also include memory cells [72], latent cells [57], and more complex models of replenishment of T cells [16,73]. However, in the present context, the two simplest models suffice.So far, we discussed untreated patients. This analysis can be extended to the more general case, for chronically infected patients who are in a quasi-steady state. The steady state will occur in untreated patients, patients on a mono-anti-retroviral therapy (mono ART is no longer in use), or a low-dose regimen with a low efficacy that does not cause the full suppression of viral replication. For example, 5 patients of 16 shown in Figure 2 were in one of the last two classes. In this situation, a quasi-steady state is preserved, and the virus still can evolve, causing progression to AIDS as we have described. In order to introduce such a low-dose ART, it suffices to replace the basic reproduction number, $R_{0,}\;$ with its effective value under ART, $R_{ART}=\left(1-\epsilon \right)\;R_{0}$, where ϵ, 0<ϵ<1,  is the efficacy of the treatment. Indeed, any existing ART affects only the infectivity parameter, p, which, in our model, includes both the number of virions per cells and their average infectivity. As a result, as long as the dose is low, so that $\epsilon <1-1/R_{0}$, the above results apply, and we obtain progression at a rate given by Equation (13), but with a smaller number of infected cells, $I_{ss},$ and $R_{ART}$ replacing $R_{0}$. The value of $I_{ss}$ enters through the argument of a large logarithm, Equation (6). Therefore, the change in the virus load even by two or three orders of magnitude (from $10^{8}$ to $10^{6}$ productively infected cells), which is possible if the immune response is turned off and Equation (8) applies, would cause only a modest delay of progression, or almost none at all if the immune response stays active (note that, in Equation (10), $I_{ss}$ does not depend on $R_{0}$). Therefore, according to our model, a suboptimal ART does not much affect the outcome. Unfortunately, this prediction was observed in early patients including the five patients in the cited data, Figure 2. Monotherapy and low-dose treatment are not in the clinical use today, because they are associated with resistant mutations and, as we just discussed, rapid progression to AIDS.

However, if triple high dose ART is successfully applied, causing the virus load drop down to very small levels, such that the immune response is turned off, the continuous steady state is destroyed. In this case, our model is no longer appropriate, because it predicts full virus clearance. Real patients show random residual viremia spikes that occur due to rare reactivation of latent cells and not 100% efficacy of the drug. To describe this scenario, we would need to introduce latent cells into the picture [57] and consider fluctuations—i.e., stochastic virus evolution occurring during the spikes. Because the spikes are short, and replication and hence evolution of virus are suppressed by the drug, a highly efficient therapy is expected to delay the onset of AIDS a lot, which is what they see in patients. This case requires a separate treatment. 

To summarize, the present work proposes a model of AIDS progression based on the virus adaptation that fits the existing data better than the homeostatic deregulation-based models. The predicted dependence of $t_{\mathrm{AIDS}}$ on the virus load is consistent with the observed slow, negative correlation between the time to AIDS and viremia. The evolution model also explains, at least qualitatively, the positive correlation observed between the speed of the progression to AIDS and the level of the immune activation. Indeed, high diversity and fast evolution of HIV is likely to be driven by the early immune escape in epitopes and the subsequent compensatory mutations [41]. Hence, the average selection coefficient correlates with the intensity of the immune response. The model also naturally explains the lack of progression to AIDS in SIV strains infecting their natural host [29,30,31,32,33], where the virus and the natural host have already co-evolved for million of years, unlike for HIV in humans.

The proposed evolution model of progression to AIDS is very easy to falsify, because it makes a testable, closed prediction for the time to AIDS based on a set of patient and virus parameters obtained from a single blood sample from an individual patient. We hope to test our model on live data elsewhere.

## Figures and Tables

**Figure 1 microorganisms-08-01714-f001:**
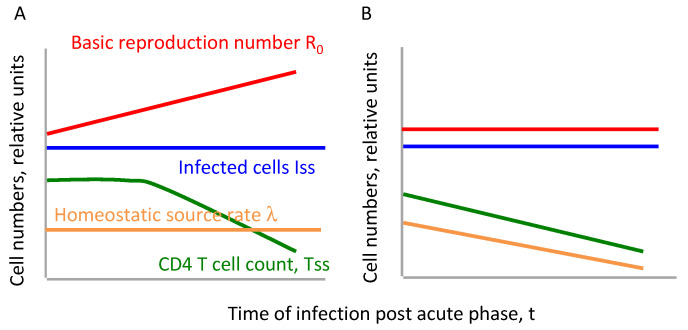
Two alternative hypotheses of the gradual decline of CD4 T cell count tested in the present article. Green line shows the quasi steady state level of CD4 T cells predicted by Equation (10) if (**A**) basic reproduction ratio $R_{0}$ changes due to gradual viral adaptation, (**B**) $R_{0}$ stays constant, but the homeostatic source of CD4 T cells becomes weaker in time.

**Figure 2 microorganisms-08-01714-f002:**
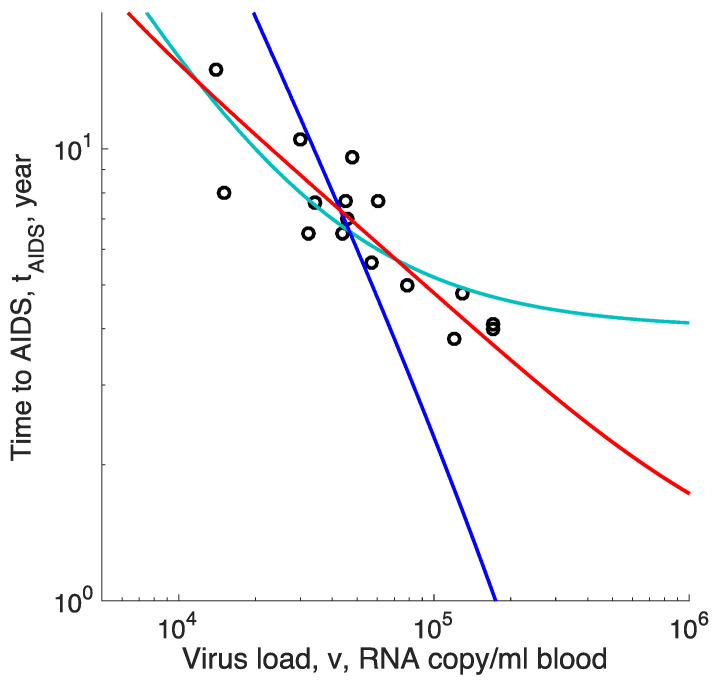
Time to AIDS predicted by three models with two composite fitting parameters compared to data isolated from HIV-positive men seroconverted during study [22]. Out of 16 patients, 11 were untreated and 5 received some form of suboptimal anti-retroviral treatment (not triple cocktails) that did not affect the outcome. Dots: Survival time versus virus load. Red curve: Best-fit prediction of the virus evolution model, Equation (14). Cyan curve: The model of a limited homeostatic source, Equation (16). Blue line: The model of the virus-induced decay of homeostatic environment, Equation (18).

**Table 1 microorganisms-08-01714-t001:** Model parameters relevant in the quasi-steady state.

Notation	Name	Units	Range
ψ	Drop in log CD4 T cell count	1	~1
$I_{ss}$	Infected cell number	cell	$100–3\times 10^{9}$
L	Total number of sites	1	10–1000
μ	Mutation rate	1/generation/nucleotide	$3\times 10^{-5}\;$ $\left(3\times 10^{-6}–10^{-4}\right)$
$U_{b}=\mu L$	Mutation rate per genome	1/generation/genome	$3\times 10^{-4}–0.03$
b	Effective selection coefficient	1	0.1–0.005
r	Recombination probability	1	0–1
T0	Number of target cells in an uninfected person	cells/mm^3^	700–1400
dT	Inverse lifespan of target cells	1/day	0.1–0.01
dI	Inverse lifespan of infected cells	1/day	0.7–1.2
R0(0)	Initial basic reproduction ratio	1	5–11
R* =T0dTIssdI	Composite parameter	1

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
