# Peer review of "An Evolutionary Model of Progression to AIDS"

_microorganisms, 2020, doi:10.3390/microorganisms8111714_

Round 1

Reviewer 1 Report

Understanding the dynamics HIV disease is essential for depicting, developing, and investigating effective treatment strategies. The concept of this manuscript is interesting to readers. And the model is well established. But there are some issues need to be addressed:

First, the author should describe more clearly regarding samples been used in this study if possible, eg. use a “table” to present patient age, gender, progress to AIDS, therapy status, viral load…

Second, the results were not described well and interpreted well, which made this manuscript difficult to understand. It would be better to present a table to describe the variable and parameter values of this model.

Third, more analysis should be performed to demonstrate the advantage and feasibility of this model. To assess the model, the author should evaluate how this model works in different situations. How different values of parameters affect progression to AIDS? Like treatment/control situation with different values of those parameter causes difference progression to AIDS.

Reviewer 2 Report

Rouzine examined, using mathematical modeling, a hypothesis of variable viral fitness during HIV infection and found that this potential mechanism could better explain the observed negative correlation between latency of viral reoccurrence and the viral set point. I think this study is interesting and the mathematical analysis is rigorous. The manuscript is well-written (with only minor typos).

I have one comment about the model fitting. The author fixed a number of model parameters which are drawn from the literature. This is sensible given that the data are probably too limited to be sufficient to infer all the model parameters. However, I would like to know how sensitive the best-fit curves are to those fixed parameters, because the conclusion may change if the fitting curves (in Figure 2) are very sensitive to the choice of those fixed parameters. Therefore, I think a sensitivity analysis is absolutely necessary to show that how the fitting curves in Figure 2 change if those fixed parameters show some level of uncertainty. The uncertainty should be found from the literature, as statistical uncertainty (e.g. 95% confident interval, standard error, or some other kinds of measure) should usually be provided along with the point estimate of a biological parameter in previous experimental or modelling studies. 

Minor changes:

line 26: "with men"?

line 36: change "is" to "was"

line 44: what does "perfect" mean?

line 47: swap "both" and "with"

line 130: "In" the more

I cannot find any detail about the method for model fitting, which should be provided for reproducibility purpose (Least-squares? how to choose the starting points? how to find the global minimum? software used to implement the fitting? etc.).

Reviewer 3 Report

All the comments are reported in the attached file "ReviewForAuthors.docx".

Round 2

Reviewer 1 Report

Thank author for effort and time on the revision. However, I think the description and explanation in this manuscript are still not good enough for publication.

As a model for prognosis, it’s very important to know what situations the model is applicable to. That’s why I asked in my comments about the patient information and the data classification. I understand that this manuscript didn’t use any samples, but it did use the patient data to compare alternative models. The rationality of the data classification and application will affect the validity of the evaluation and comparison. The scope of this model is not clear in this manuscript and is still not interpreted well.

In the old version, this manuscript demonstrate that they develop a mechanistic mathematical model to predict the speed of progression to AIDS in both treated and untreated patients. In this revised version, the author deleted the aim group: “treated patient”. But when fitting data, the author used data from 16 patients progressed to AIDS. Of those 16 patients, 5 patients received antiviral treatment. No analysis about this data explaining on treated patients. The conclusion is “This method can be used for patient prognosis, if we know the input parameters in Eq.13”. This conclusion doesn’t define the scope of this model (only “untreated patients” or for both “treated and untreated patients”?).

Author Response

Reviewer:

As a model for prognosis, it’s very important to know what situations the model is applicable to. That’s why I asked in my comments about the patient information and the data classification. I understand that this manuscript didn’t use any samples, but it did use the patient data to compare alternative models. The rationality of the data classification and application will affect the validity of the evaluation and comparison. The scope of this model is not clear in this manuscript and is still not interpreted well.
In the old version, this manuscript demonstrate that they develop a mechanistic mathematical model to predict the speed of progression to AIDS in both treated and untreated patients. In this revised version, the author deleted the aim group: “treated patient”. But when fitting data, the author used data from 16 patients progressed to AIDS. Of those 16 patients, 5 patients received antiviral treatment. No analysis about this data explaining on treated patients. The conclusion is “This method can be used for patient prognosis, if we know the input parameters in Eq.13”. This conclusion doesn’t define the scope of this model (only “untreated patients” or for both “treated and untreated patients”?).

Response:

Thank you for the comment. I made changes, as follows:

1) Phrase

"and patients treated with suboptimal therapy"

is added to the Abstract.

2) Text in Fig. 2 caption now reads:

"Time to AIDS predicted by three models with two composite fitting parameters compared to data isolated from HIV-positive men seroconverted during study [22]. Out of 16 patients, 11 were untreated and 5 received some form of suboptimal anti-retroviral treatment (not triple cocktails) that did not affect the outcome."

3) At the end of Discussion, I added a paragraph discussing the extension of the results to the case of patients who received suboptimal therapy, like the 5 patients in Fig.2. I explain that the model does not apply in the case of successful HAART.

Reviewer 2 Report

My comments have been addressed. No further comments from me.

Author Response

No changes required.
